# DNA Vaccine Co-Expressing Hemagglutinin and IFN-γ Provides Partial Protection to Ferrets against Lethal Challenge with Canine Distemper Virus

**DOI:** 10.3390/v15091873

**Published:** 2023-09-04

**Authors:** Jianjun Zhao, Yiyang Sun, Ping Sui, Hongjun Pan, Yijun Shi, Jie Chen, Hailing Zhang, Xiaolong Wang, Rongshan Tao, Mengjia Liu, Dongbo Sun, Jiasan Zheng

**Affiliations:** 1Key Laboratory of Bovine Disease Control in Northeast China, Ministry of Agriculture and Rural Affairs, College of Animal Science and Veterinary Medicine, Heilongjiang Bayi Agricultural University, Daqing 163319, Chinadongbosun@126.com (D.S.);; 2Institute of Special Animal and Plant Sciences, Chinese Academy of Agricultural Sciences (CAAS), Changchun 130112, China15134075643@163.com (J.C.);; 3Yantai Animal Disease Control Center of Shandong Province, Yantai 264000, China; 4Agricultural Bureau of Shanyang Country, Shangluo 726400, China; 5The Key Laboratory of Environmental Pollution Monitoring and Disease Control, School of Public Health, Guizhou Medical University, Guiyang 550025, China; 6Jinan Customs in Shandong Province of the P.R. of China, Jinan 250000, China

**Keywords:** canine distemper virus, DNA vaccine, haemagglutinin, IFN-γ, ferret

## Abstract

Canine distemper (CD), caused by canine distemper virus (CDV), is a highly contagious and lethal disease in domestic and wild carnivores. Although CDV live-attenuated vaccines have reduced the incidence of CD worldwide, low levels of protection are achieved in the presence of maternal antibodies in juvenile animals. Moreover, live-attenuated CDV vaccines may retain residual virulence in highly susceptible species and cause disease. Here, we generated several CDV DNA vaccine candidates based on the biscistronic vector (pIRES) co-expressing virus wild-type or codon-optimized hemagglutinin (H) and nucleocapsid (N) or ferret interferon (IFN)-γ, as a molecular adjuvant, respectively. Apparently, ferret (*Mustela putorius furo*)-specific codon optimization increased the expression of CDV H and N proteins. A ferret model of CDV was used to evaluate the protective immune response of the DNA vaccines. The results of the vaccinated ferrets showed that the DNA vaccine co-expressing the genes of codon-optimized H and ferret IFN-γ (poptiH-IRES-IFN) elicited the highest anti-CDV serum-neutralizing antibodies titer (1:14) and cytokine responses (upregulated TNF-α, IL-4, IL-2, and IFN-γ expression) after the third immunization. Following vaccination, the animals were challenged with a lethal CDV 5804Pe/H strain with a dose of 10^5.0^ TCID_50_. Protective immune responses induced by the DNA vaccine alleviated clinical symptoms and pathological changes in CDV-infected ferrets. However, it cannot completely prevent virus replication and viremia in vivo as well as virus shedding due to the limited neutralizing antibody level, which eventually contributed to a survival rate of 75% (3/4) against CDV infection. Therefore, the improved strategies for the present DNA vaccines should be taken into consideration to develop more protective immunity, which includes increasing antigen expression or alternative delivery routes, such as gene gun injection.

## 1. Introduction

Canine distemper (CD), caused by canine distemper virus (CDV), is a highly contagious disease with typical acute systemic clinical symptoms, including fever, rash, viremia, and central nervous system damage [1,2], and it is associated with an increased risk of secondary infection [3]. In highly susceptible natural hosts, such as mink and ferrets, CD is often lethal [4].

While the use of CDV live-attenuated vaccines has markedly reduced the incidence of CD worldwide [5], low levels of protection have been reported in the presence of maternal antibodies [6]. Moreover, attenuated CDV vaccine strains may cause immunosuppression or neurological complications in susceptible hosts [7], and residual virulence may become lethal when applied in wildlife [8].

To overcome the limitations mentioned above, DNA vaccines have been developed. They contain no infectious components, stimulate both long-lasting cellular and humoral immune responses without any risk of reversion to virulence, and are not affected by maternal antibodies [8,9,10]. A DNA vaccine based on the hemagglutinin (H) gene of a CDV wild-type strain was reported to induce a higher level of anti-CDV-neutralizing antibodies (NAb; 1:64) in mice, compared with the fusion (F) gene (1:16) [8]. Furthermore, a DNA vaccine derived from the H gene of the CDV vaccine Onderstepoort strain can effectively induce humoral immunity in young animals (5-day-old minks), with an anti-CDV NAb titer reaching 1:100 after three immunizations, which is considered sufficient to protect minks from lethal CDV challenge [9]. A recent study showed that DNA vaccines combining H, F, and/or nucleocapsid (N) plasmids were no more effective in ferrets than those carrying the H plasmid alone [10]. Although CDV N protein cannot effectively induce anti-CDV NAb, it does induce cytotoxic lymphocyte (CTL) reactions following a second inoculation [10]. 

Codon optimization has been proven to be an effective strategy for improving the immune effect of DNA vaccines by increasing antigen expression levels [11]. Grote et al. [12] demonstrated that substitution with the codon most preferred by the host, without altering the encoded amino acids, increases protein expression in host cells. Moreover, the incorporation of interferon-gamma (IFN-γ) in the DNA vaccine, which exhibits immunomodulatory functions, can improve the humoral and cellular immunity induced by vaccines [13]. In a study evaluating the use of IFN-γ as an adjuvant for an HIV-1 DNA vaccine in mice, it was observed that this approach resulted in increased antigen-specific antibody responses and significantly enhanced proliferation of antigen-specific helper T_H_ cells [14].

In the present study, we developed CDV DNA vaccines based on the bicistronic pIRES vector, which harbored codon optimization target antigens (optiH and/or optiN) and co-expressed ferret IFN-γ as a molecular adjuvant. The DNA vaccines were then evaluated in ferrets for induced humoral and cellular immune responses as well as protective responses against challenge with a ferret-adapted CDV 5804Pe/H strain.

## 2. Materials and Methods

### 2.1. Viruses, Cells, and Ferrets

CDV ferret-adapted wild-type strain 5804Pe/H was gifted by Professor Veronica von Messling (Institute of Paul Ehrlich Institute, Langen 63225, Germany) and propagated in VeroDogSLAM cells [15,16]. CDV SD(14)7 strain isolated from a infected breeding arctic fox (*Vulpes lagopus*) in China was propagated in VeroDogSLAM cells [17]. The BHK-21 cells used for indirect immunofluorescence assay were cultured at 37 °C with 5% CO_2_ in Dulbecco’s minimum essential medium (DMEM) supplemented with 10% fetal calf serum.

Twenty ferrets (*Mustela putorius furo*), comprising ten males and ten females (2–3 months old), each weighing 400–500 g, were purchased from WuXi Shanhujiao Biotechnology Co., Ltd., Wuxi 214000, Jiangsu Province, China. All the animals were healthy and tested sera antibody and antigen negative for CDV.

### 2.2. Preparation of DNA Vaccines

To obtain the CDV H and N genes, which are the target genes of the DNA vaccine, viral RNA was extracted from CDV strain SD(14)7, and reverse transcription–polymerase chain reaction (RT-PCR) was performed. The H and N genes were subjected to codon optimization based on the ferret species using an online codon optimization tool (http://molbio.info.nih.gov/dnaworks/, accessed on 1 October 2015.), and the genes were synthesized by Shanghai Jierui Bioengineering Co., Ltd., Shanghai 200000, China. The sequence alignment of the parental CDV gene and the codon-optimized gene is included in (Appendix A. The ferret IFN-γ gene was obtained from the recombinant plasmid expressing ferret IFN-γ via PCR [18] (Table A1).

Restriction enzyme *Not*I sites were inserted at the 5′ ends and *Sal*I at the 3′ ends of the cDNA encoding the N and IFN-γ genes. *Nhe*I sites were inserted at the 5′ ends and *Xho*I sites at the 3′ ends of the cDNA encoding the H gene. The target genes were sequentially cloned into the eukaryotic expression pIRES vector; then, five recombinant plasmids, pH-IRES (H), pH-IRES-N (H-IRES-N), poptiH-IRES (optiH), poptiH-IRES-optiN (optiH-IRES-optiN), and poptiH-IRES-IFN-γ (optiH-IRES-IFN), were constructed (Figure 1). The recombinant plasmids were purified using the Genopure Plasmid Midi Kit (Roche, Kaiseraugst 4303, Switzerland).

### 2.3. Identification of DNA Vaccine Plasmids via Indirect Immunofluorescence Assay (IFA)

BHK-21 cells were seeded into a 6-well plate and cultured at 37 °C with 5% CO_2_. A cell monolayer at 80% confluence was transfected with either H, optiH, H-IRES-N, optiH-IRES-optiN, optiH-IRES-IFN, or an empty plasmid vector, pIRES, using Lipofectamine 2000 (Thermo Fisher, Carlsbad, CA 92010, USA) according to the manufacturer’s instructions. At 48 h after transfection, the cells were fixed in 1 mL of 4% formaldehyde for 30 min at 37 °C and permeabilized in 1 mL of 0.5% Triton X-100 for 10 min. After three washes with phosphate-buffered saline (PBS), samples were blocked with goat serum for 30 min at 37 °C. The samples were washed and blocked after each subsequent antibody exposure to anti-CDV H protein monoclonal antibody (1C42H11 monoclonal antibody, VMRD, Pullman, WA, 99163, USA), anti-CDV N protein monoclonal antibody (CDV-NP monoclonal antibody, VMRD, USA), and anti-mink IFN-γ monoclonal antibody [19], as appropriate for the corresponding plasmid vectors, whereas goat anti-mouse IgG H-L (FITC) (Abcam, Cambridge 02139, UK) was used as the secondary antibody. Images were obtained using a fluorescence microscope (EVOS FL, Thermo, USA). Five fields of vision for each image were randomly selected, and the mean number of fluorescent cells was calculated and compared to evaluate protein expression levels.

### 2.4. Administration of DNA Vaccines and Sample Collection

Twenty ferrets were equally divided into five groups (*n* = 4/group) and were immunized with optiH, optiH-IRES-IFN, optiH-IRES-optiN, H, and control (pIRES) plasmids, respectively. The ferrets were vaccinated thrice at 3-week intervals (Figure 2). Immunization was performed under anesthesia via intramuscular injection of 10 mg/kg of Zoletil-50 (Virbac). Specific DNA vaccine immunization plans are shown (Table 1). The vaccine doses administered for each animal were 800 μg, 800 μg, and 500 μg of each plasmid in the first, second, and third immunization, respectively [8,9,10]. One-third of the dose was administered intradermally; the remaining dose was administered intramuscularly. Whole blood samples were obtained from the jugular veins of ferrets before and after each immunization (0,3, 6, and 9 weeks). The sera were separated from the whole blood and stored at −20 °C until they were used in virus neutralization and cytokine assays.

Three weeks after the last immunization, the 20 ferrets were divided into five groups (*n* = 4/group), then separately housed and inoculated nasally with 10^5.0^ TCID_50_ of CDV strain 5804Pe/H. Clinical symptoms, rectal temperature, and weight changes were monitored daily throughout the 21-day post-challenge (dpc) period, as well as 9, 10, 11, and 12 weeks after CDV challenge (Figure 2). The sera were separated from the whole blood and stored at −20 °C until they were used in virus neutralization and cytokine assays. Nasal and rectal swabs were collected at 0, 3, 6, 9, 12, 15, 18, and 21 dpc, suspended in 0.5 mL of sterile PBS, and stored at −20 °C until they were used for CDV RNA extraction. Tissue samples (lung, spleen, lymph, liver, intestines, and brain) were collected at 21 dpc and stored at −20 °C.

The study protocol was approved by the Institute of Special Economic Animal and Plant Sciences, CAAS, and was performed in accordance with animal ethics guidelines and approved protocols (Animal Ethics Committee Approval Number: ISAPSAEC-2019-003D). Animal welfare was monitored daily. All animals are euthanized with an overdose of Zoletil-50 (50 mg/kg, Virbac) when they reached the humane endpoint criteria or the experimental endpoint (21 dpc). Humane endpoint criteria were defined as follows: animal does not eat or drink anymore, >20% loss of body weight, moderate to serious circulation problems or breathing difficulties, or moderate to serious behavioral display of clinical symptoms.

### 2.5. Assessment of Clinical Symptoms

Clinical symptoms were graded on a three-point grading scale (0–2) and included rectal temperature, weight, and conjunctivitis (Table 2). The parameters were analyzed as described previously with minor modifications [20]. The final score was calculated by adding the values determined for each parameter in each animal and dividing the total by the number of animals in each group.

### 2.6. Virus Neutralization and Cytokine Detection

NAbs responses in the sera of infected ferrets were determined using a neutralization assay in VeroDogSLAM cells. Briefly, duplicate two-fold dilutions (starting at 1:2) of serum samples (heat-inactivated at 56 °C) were incubated in triplicate with 200 TCID_50_ of 5804Pe/H for 1 h at 37 °C. Subsequently, trypsinized VeroDogSLAM cells were added at a concentration of 1 × 10^4^ cells/well. The plates were incubated for 5–7 days at 37 °C and visually monitored for cytopathic effects. The NAb titers were calculated using the method described by Reed and Muench [21].

To further evaluate the ability of the vaccine to induce immune cell-mediated responses, canine interleukin-2 (IL-2), tumor necrosis factor (TNF-α), interleukin-4 (IL-4), and IFN-γ enzyme-linked immunosorbent assay (ELISA) kits (R&D Systems, USA) were used to detect the cytokines present in the sera of immunized or infected ferrets, following the manufacturer’s instructions.

### 2.7. Real-Time RT-PCR Detected CDV RNA

RNA extraction was performed following the manufacturer’s instructions. The clinical suspensions from the nasal and rectal swabs were subjected to viral RNA extraction using the QIAamp Viral RNA Kit (Qiagen, Hilden 40724, Germany). Total RNA was extracted from the tissue samples using the RNeasy Mini Kit (Qiagen, Germany). The QIAamp Blood RNA Mini Kit (Qiagen, Germany) was used to extract CDV RNA from blood samples. Viral RNA was reverse-transcribed using random hexamer primers and the Prime Script II 1st Strand cDNA Kit (Takara, Gunma 3700344, Japan). CDV real-time RT-PCR was performed in a LightCycler^®^96 (Roche, Switzerland), as previously described [22].

### 2.8. Histology and Immunohistochemistry

Histological examinations were performed on all infected animals. The tissues were fixed in 10% formaldehyde in PBS and processed for histopathology. That is, paraffin-embedded tissue sections (lung, spleen, and brain) were obtained and stained with hematoxylin and eosin. Immunohistochemical analysis was performed on paraffin-embedded sections using mouse CDV-NP monoclonal antibody (VMRD, USA) and the EnVison/HRP Systems kit (Dako, Glostrup 2600, Denmark), as previously described [23].

### 2.9. Statistical Analysis

Statistical analysis was performed using GraphPad Prism software, version 9.0. All data were expressed as mean ± standard error of the mean (SEM) and analyzed using one-way analysis of variance (ANOVA). The differences between groups were examined for statistical significance using Student’s *t*-test. A *p*-value < 0.05 was considered statistically significant.

## 3. Results

### 3.1. Codon Optimization Enhances the Expression of H and N Proteins

The expression of CDV H and N proteins and IFN-γ was detected via indirect immunofluorescence assay 48 h after transfection of the plasmids into BHK-21 cells. For all monovalent plasmids, H protein expression was observed in the cells (Figure 3A,B). Similarly, when transfected with multivalent plasmids, the BHK-21 cells expressed both antigens (Figure 3C–F,H). As expected, the expression level of H and N proteins was improved because of the genetic codon optimization. That is, the count of expressed positive cells transfected with optiH was approximately 4.5-fold higher than that of cells transfected with H (*p* < 0.001; Figure 3A,B, and Figure A1A). Similarly, for multi-valent plasmid transfection, the count of positive cells in the optiH-IRES-IFN group, regardless of whether H or N protein was detected, was significantly higher than that in the corresponding wild-type group (*p* < 0.05; Figure 3C–F and Figure A1B,C).

### 3.2. Virus-Neutralizing Antibody Responses after DNA Vaccine Immunization

Unfortunately, one ferret (#20) died from whole blood collection on day 7. Thus, 19 animals were included in follow-up analysis. Due to the limited number of ferrets, we only evaluated four recombinant DNA vaccines, excluding the vaccine H-IRES-N.

To gain insights into the level of humoral immunity induced by the DNA vaccines, the anti-CDV serum average NAbs kinetics was determined after immunization and challenge. NAbs can be detected in the optiH and optiH-IRES-IFN groups 3 weeks after the first immunization, while the second immunization (6 weeks) was required before the NAb of the H and optiH-IRES-optiN groups reached to detectable levels. After the third immunization (9 weeks), all the immunized groups induced humoral immunity and showed seroconversion with NAb titers above 1:8. The average NAb titers reached up to 1:100 after 3 weeks post-challenge (12 weeks) in immunized groups, except the H-N group. Interestingly, the optiH-IRES-IFN-immunized animals maintained the highest average NAb levels during the experimental period among the four DNA vaccine groups. In fact, after the third inoculation, the NAb levels of the optiH-IRES-IFN group were significantly higher than those of the H group (14 VS 9; *p* < 0.01), and the NAb level quickly reached up to 1:211 at 3 weeks after CDV challenge, which was significantly higher than that of the optiH-IRES-optiN group (211 VS 47; *p* < 0.05) (Figure 4).

### 3.3. Cytokine Responses in Sera after Immunization with DNA Vaccines

DNA vaccines promoted the expression of TNF-α, IL-4, IL-2, and IFN-γ in the sera of immunized ferrets. The upward trend was observed for all four cytokines over time, with levels stabilizing after the third immunization. In contrast, no significant fluctuation was observed in the control ferrets. The IFN-γ expression level in the optiH-IFN group peaked 3 weeks after the second immunization (2.83 pg/mL; Figure 5A) and was significantly higher than that of the control group (1.7 pg/mL; *p* < 0.05). A high level of IFN-γ was detected via ELISA in optiH-IRES-IFN at 7 dpc (2.5 pg/mL). Interestingly, the levels of TNF-α, IL-2, and IL-4 induced by optiH-IRES-IFN were also higher than those in the control group (Figure 5B–D). In particular, the levels of IL-2 (*p* < 0.001) and IL-4 (*p* < 0.0001) secretion differed significantly from those of the control group at 7 dpc (Figure 5C).

Further, 21 days (9 weeks) after the third immunization, the level of TNF-α was also significantly increased in the optiH-IRES-IFN group (1.7 pg/mL) compared with the control group (0.9 pg/mL; *p* < 0.05) (Figure 5B). The expression level of IL-4 (4.2 pg/mL) was also significantly higher than that of the control group (2.2 pg/mL; *p* < 0.0001) and that in the H group (3.5 pg/mL; *p* < 0.05) at 7 dpc (Figure 5D).

The levels of all four cytokines in the vaccine groups remained higher than those in the control group throughout the entire study period, with the optiH-IRES-IFN group exhibiting the strongest induction effect. In contrast, the induced levels of cytokines in the H group were relatively weaker than that in the other immune groups; however, the induced IL-4 level (3.5 pg/mL) was still significantly higher than that in the control group (2.2 pg/mL) at 7 dpc (Figure 5D). Meanwhile, the ferrets vaccinated with optiH or optiH-IRES-optiN showed similar trends, with no significant differences in cytokine expression after immunization and challenge.

### 3.4. Vaccine Immunization Alleviates Clinical Symptoms in Infected Ferrets

Twenty-one days after the third immunization, a total of 19 ferrets were inoculated intranasally with 10^5.0^ TCID_50_ of the CDV strain 5804Pe/H. After a 3-day incubation period, all three control ferrets developed similar fever (>39 °C), with symptoms of depression, weight loss, and purulent conjunctivitis, and they succumbed to infection within 14 dpc (Figure 6A–C). In contrast, DNA-vaccine-immunized ferrets, except the ferrets in the H group showing moderate CD symptoms, showed transient fever in 3–5 dpc, then quickly recovered with limited body weight fluctuation and mild conjunctivitis (Figure 6A,B). Particularly, although the DNA vaccines cannot provide 100% protection to ferrets against lethal challenge, the lower mortality rates were observed in four immunized groups (optiH (1/4), optiH-IRES-IFN (1/4), optiH-IRES-optiN (1/4), and H (2/4)) when compared with the pIRES control group (3/3) (Figure 6C).

Furthermore, a clinical symptom scoring system was established to score and evaluate the immunized effects of different vaccine groups (Figure 7). Immune group optiH showed the lowest score (0.58), followed by the optiH-IRES-IFN (0.67), optiH-IRES-optiN (0.83), and H groups (0.83), while the control group showed the highest score (1.89). The results of clinical symptom scoring agreed with the anti-CDV NAb responses after DNA vaccine immunization, suggesting that DNA vaccine immunization reduced the severity of clinical symptoms in infected animals.

### 3.5. Vaccine Immunization Reduces CDV RNA Loads in Infected Ferrets

Viral release dynamics were measured via viral RNA loads in the nasal and rectal swabs of infected animals at various time points (Figure 8A,B). The release of the virus over time in the bodies of infected ferrets was reflected by the viral RNA loads detected in nasal swabs. Viral RNA was first detected 3 dpc, followed by a gradual increase in viral RNA load over time, peaking between 10 and 15 dpc. Viral release occurred more readily via the rectal route than the nasal route. Peak viral release corresponded to the period with the most severe clinical symptoms, during which all control animals died. In contrast, the viral RNA loads decreased in the immunized group after the peak viral release period, corresponding to a gradual decrease in clinical symptoms. The viral RNA loads in the control group remained significantly higher than those in the immunized group from 3 dpc (*p* < 0.01). Although the ability of the H group to prevent viral replication was slightly weaker than that of the other immunized group, significant differences were detected in the nasal swabs compared with the control group at 7 dpc and 10 dpc (*p* < 0.05) (Figure 8A). The trend in viral RNA levels within the rectum was like that in the respiratory tract, with a significant difference observed between the control group (10^6.1^ copies/mL) and optiH group (10^3.0^ copies/mL) at 10 dpc (Figure 8B). Moreover, the viral RNA loads of rectal swabs in the optiH (10^4.0^ copies/mL, *p* < 0.01) and H (10^4.1^ copies/mL, *p* < 0.05) groups peaked at 14 dpc and differed significantly from that in the control group (Figure 8B).

At the end of the experiment (21 dpc), the remaining animals (H, *n* = 2; optiH, *n* = 3; optiH-IRES-optiN, *n* = 3; optiH-IRES-IFN, *n* = 3; pIRES, *n* = 0) were euthanized for autopsy, and the viral RNA loads of each tissue were determined. The optiH and optiH-IRES-IFN groups exhibited lower viral loads in all tissues, while the optiH-IRES-IFN group had the lowest viral load in the brain (10^3.7^ copies/mL) and liver tissues (10^2.5^ copies/mL) among the immunized groups. In contrast, the optiH-IRES-optiN group had higher viral RNA loads compared with the other vaccine groups, particularly in the spleen, lymph, and other immune organs that are susceptible to CD (Figure 8C), corresponding to lower neutralizing antibody titers after the third immunization.

### 3.6. Pathological Changes and Immunohistochemistry of Ferrets after CDV Challenge

Histopathological examination revealed systemic infection in the ferrets. Typical interstitial pneumonia was observed in the control group. The necrotic bronchiolar epithelial cells were partially exfoliated, and red inclusion bodies were observed in the cytoplasm of epithelial cells (Figure A2B). Some of the nerve cells in the brain are severely swollen and vacuum-like (Figure A2D). However, ferrets immunized with the recombinant plasmid optiH-IRES-IFN only exhibited mild brain injury, splenic congestion, and interstitial pneumonia, corresponding to mild or moderate histological changes (Figure A2). Meanwhile, CDV antigens were generally identified in the tracheobronchial epithelial cells, follicles, and monocytes of the lung and spleen sections in the controls (Figure A3). Immunohistochemistry staining of the brain sections highlighted the presence of infected neurons (Figure A3C,D). It is worth noting that, even in ferrets immunized with the optiH-IRES-IFN plasmid, which was previously considered to elicit the best immune response, CDV-positive cells were also detected (Figure A3A,C,E).

## 4. Discussion

In the present study, we developed CDV DNA vaccines based on the H protein, which concurrently elicit humoral and cell-medicated immunity. The pIRES vector was used, which can co-express two target genes, the H gene and N gene, via the internal ribosomal entry site (IRES) [24]. In addition, the ferret IFN-γ gene was included as an adjuvant to promote immune responses and was cloned into the vector to co-express with the other genes. IFN-γ and CDV H protein can be synthesized in the same transfected cell and presented together by antigen-presenting cells (APCs), thereby strengthening the ability to induce an immune response.

As expected, codon optimization improves the expression of H and N proteins, facilitating the effective induction of humoral and cytokine responses by all DNA vaccines. Among the recombinant plasmids, optiH-IRES-IFN exhibited a superior neutralizing antibody induction effect, with the anti-CDV NAb titer reaching 1:210 at 21 dpc. Given that commercial ferret cytokine ELISA kits are not currently available, we employed canine ELISA kits, which are closely related to ferrets. Indeed, the associated results confirmed that changes in ferret cytokine expression can be accurately assessed using canine ELISA kits. The group immunized with optiH-IRES-IFN consistently maintained higher IFN-γ levels than the other immunization groups. After three immunizations, the concentration of IFN-γ reached 2.83 pg/mL, which was 1.5-times that of the optiH group, suggesting that the addition of the IFN-γ gene to the vaccine construct effectively increased IFN-γ secretion. We also compared the levels of TNF-α, IL-4, and IL-2 and found that those induced by optiH-IRES-IFN were consistently higher than those of other groups, indicating that the addition of an IFN-γ adjuvant not only promoted humoral immune responses but also cytokine responses. The similarity in the levels of NAb induced by optiH and optiH-IRES-optiN is due to the lack of neutralizing activity of the N gene-induced antibodies, which is consistent with the findings of previous studies [10]. In an earlier study, a DNA vaccine constructed with the N gene triggered a significant CTL (cytotoxic T cell) response after three immunizations [8]. Although we did not measure CTL response directly, we measured cytokines (such as IFN-γ, TNF-α, IL-4, and IL-2) closely related to cellular immunity in our study. Unfortunately, our experimental results show that there is no significant statistical difference in cytokine levels induced by the optiH-IRES-optiN vaccine compared with other vaccine groups. This is because the N gene is located downstream of the IRES in the plasmid, which may cause its expression level to be low and may lead to no obvious differences in cytokine levels.

Currently, the incomplete protection afforded by the CDV attenuated vaccine may be partly due to the significant genetic variations between vaccine strains and wild-type viruses of CDV [25,26]. Thus, we constructed the DNA vaccines based on the H gene of the CDV wild-type SD(14)7 strain (Asia-1 genotype). Our results showed that all the DNA vaccines were able to trigger an immune response, showing cross-protection against the lethal CDV strain 5804Pe/H (Arctic genotype), a ferret-adapted lethal CDV strain.

However, it is worth pointing out that our vaccine did not induce a satisfactory immune response as expected. After three immunizations, the highest titer of neutralizing antibody was as low as 1:14. Although the titer of neutralizing antibody in three immune groups (optiH, optiH-IRES-IFN, H) exceeded 1:100 after CDV challenge, these vaccines provide only partial protective immunity, sparing 75% of animals from death. It is important to note that vaccines cannot completely protect animals from sickness or block the shedding of viruses. Further pathological sections and immunohistochemical investigation showed that, even in the vaccine group with the best immune effect, lesions still existed. In particular, we observed that, even in the best case of the vaccine effect, the virus was continuously released in the respiratory tract and digestive tract, although the RNA load was lower than that in the control group. This is probably because ferrets showed high sensitivity to the CDV strain 5804Pe/H adapted to ferrets [15], which can also be confirmed by the total death of our control group within 14 days after challenge. Furthermore, this occurrence could potentially be linked to the inadequate level of antigen expression, as substantiated by the findings from indirect immunofluorescence. Changing to more efficient expression vectors may solve this problem.

We attempted to achieve high vaccine doses by administering one-third intradermal injection and two-thirds intramuscular injection, which has been shown to effectively activate Langerhans cells and muscle cells, inducing high levels of antibodies [27]. However, DNA vaccines are typically hindered by the instability of the plasmid superhelix structure in vivo, which can be countered using a liposome adjuvant. Liposomes can promote the maturation of dendritic cells, protect antigens, and prevent their degradation in vivo [28]. After wrapping the antigens, the liposomes take on a slow-release role, thus enabling the body to acquire long-term immunity [29]. In addition, the selection of improved delivery methods can enhance immunity, such as using microneedles [30], low-frequency ultrasound as a transcutaneous immunization adjuvant [31], or electroporation [32].

Although the immune effect in our study was not as effective as expected, DNA vaccines are safe for the CD immunization of wildlife. Enhancing the protective immune response by increasing antigen expression and improving antigen delivery levels would be a future research direction for CDV DNA vaccines.

## 5. Conclusions

We constructed CDV DNA vaccines based on the bicistronic pIRES vector, co-expressing wild-type CDV H or codon-optimized H, as well as N protein or ferret IFN-γ. These plasmids effectively induced neutralizing antibody responses against CDV and cytokine expression in a ferret model. Our results indicate that co-expression of an IFN-γ molecular adjuvant can improve the levels of neutralizing antibodies and cytokines induced by our DNA vaccine. Collectively, this vaccine was found to provide partial protection against CDV infection; however, antigen expression and vaccine delivery mode require further optimization to achieve higher protection efficiency.

## Figures and Tables

**Figure 1 viruses-15-01873-f001:**
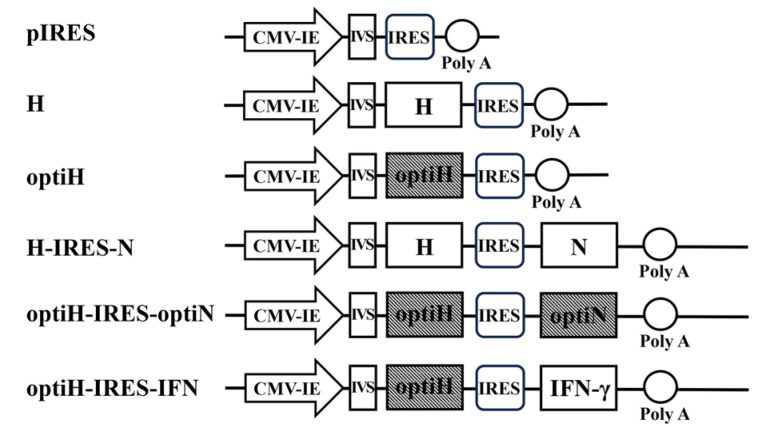
Schematic representation of the plasmids used for DNA immunization. The structural compositions of H, H-IRES-N, optiH, optiH-IRES-optiN, optiH-IRES-IFN, and pIRES are shown.

**Figure 2 viruses-15-01873-f002:**
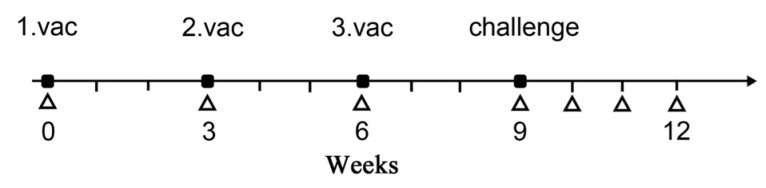
Ferret immunization, challenge, and blood sampling schematic. The ferrets were triple vaccinated with DNA vaccine H (*n* = 4), optiH (*n* = 4), optiH-IRES-IFN (*n* = 4), optiH-IRES-optiN (*n* = 4), or pIRES (*n* = 4). All ferrets were challenged with 5804Pe/H 3 weeks after the last immunization. Triangles indicate the time of whole blood collection.

**Figure 3 viruses-15-01873-f003:**
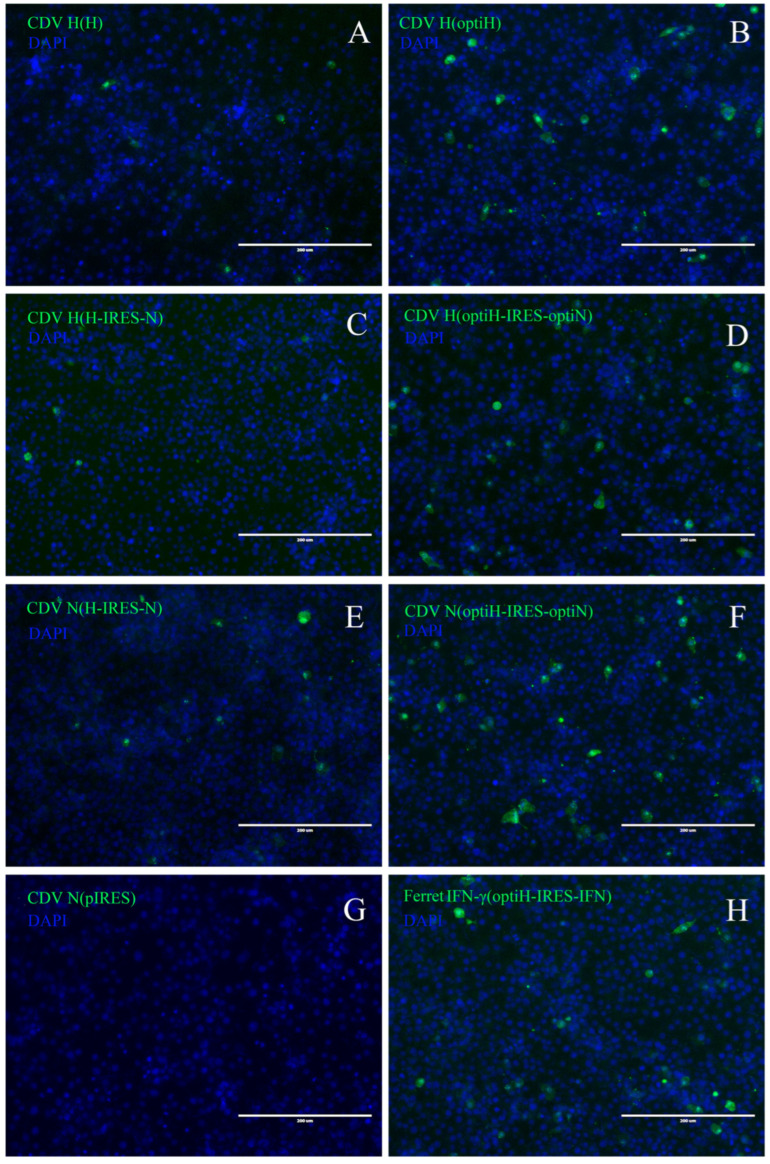
Antigen expression was detected using indirect immunofluorescence. The BHK-21 cells were transfected with one of the recombinant plasmids (H, H-IRES-N, optiH, optiH-IRES-IFN, optiH-IRES-optiN, and pIRES). Indirect immunofluorescence was performed 48 h after plasmid transfection. Bar = 200 µm. H (**A**), optiH (**B**), H-IRES-N (**C**), and optiH-IRES-optiN (**D**) were detected using a monoclonal antibody against the CDV H protein (1C42H11). H-IRES-N (**E**), optiH-IRES-optiN (**F**), and pIRES (**G**) were detected using a mouse monoclonal antibody against CDV-NP. optiH-IRES-IFN (**H**) was detected using monoclonal antibodies against mink IFN-γ.

**Figure 4 viruses-15-01873-f004:**
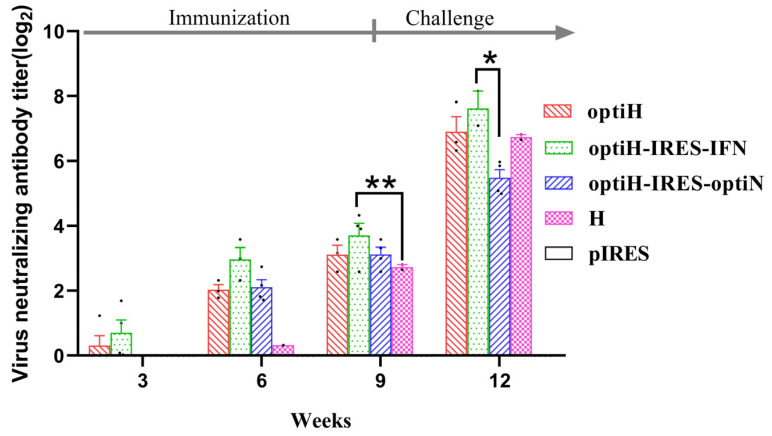
Comparison of the anti-CDV NAb levels in ferrets after immunization with different DNA vaccines and virus challenge. Blood samples were collected at the indicated time points. The results are presented as mean ± SEM. (* *p* < 0.05, ** *p* < 0.01).

**Figure 5 viruses-15-01873-f005:**
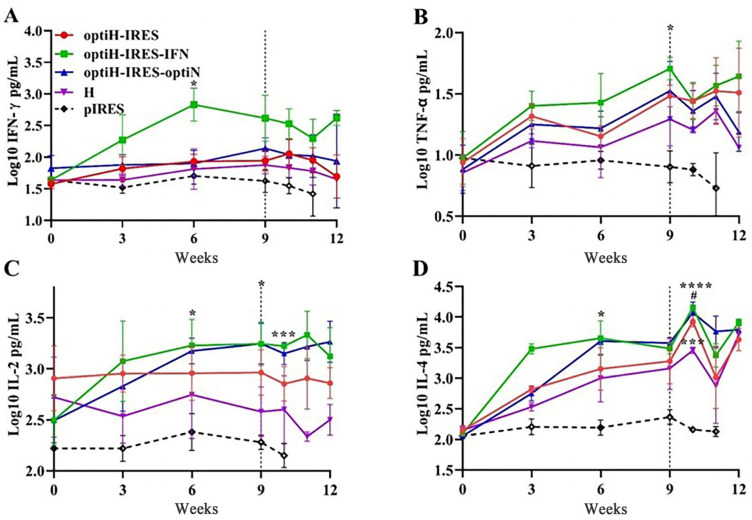
Comparison of the cytokine levels of IFN-γ (**A**), TNF-α (**B**), IL-2 (**C**), and IL-4 (**D**) after immunization and challenge. Vertical dashed lines represent the time of CDV challenge. * and # indicate a statistically significant difference (* *p* < 0.05, *** *p* < 0.001, **** *p* < 0.0001, compared with the control group; # *p* < 0.05, compared with H).

**Figure 6 viruses-15-01873-f006:**
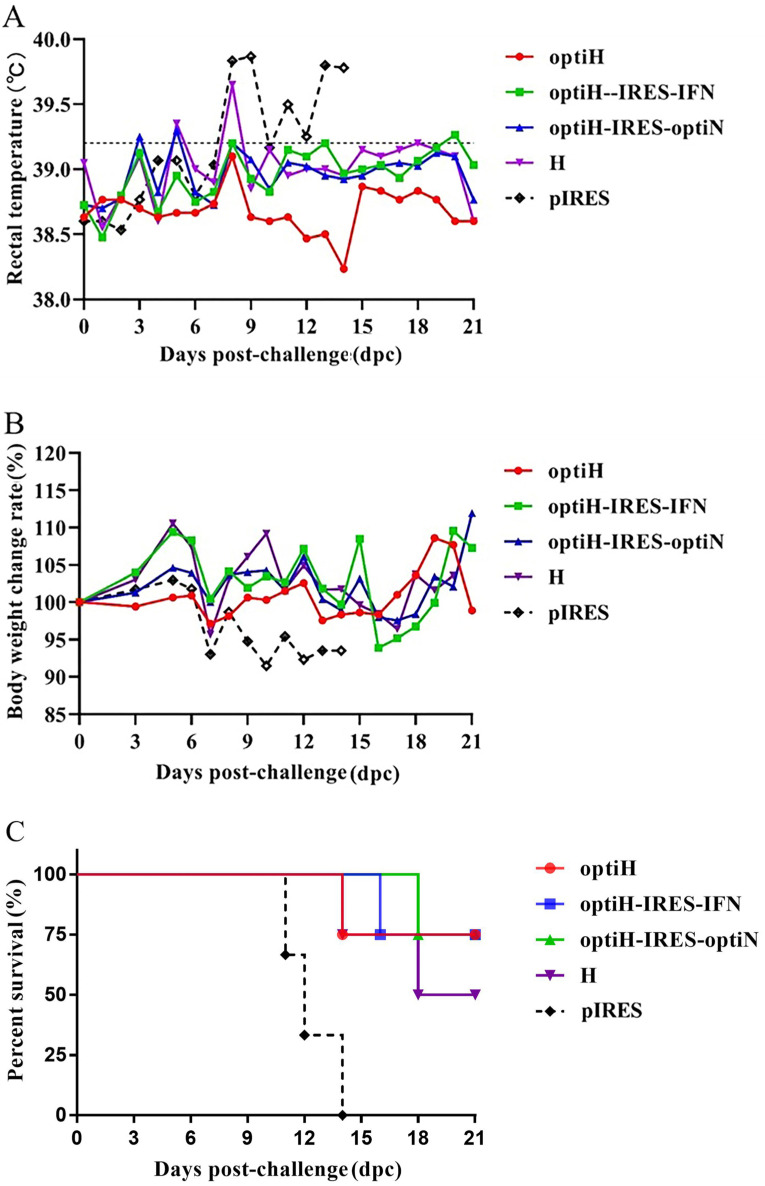
Clinical symptoms and disease course after challenge. (**A**) Rectal temperature after challenge with 5804Pe/H after immunization with different recombinant plasmids. Temperatures were recorded daily. The dashed line indicates the temperature considered to be febrile (39.2 °C). (**B**) Daily changes in body weight after the challenge. (**C**) Kaplan–Meier survival analysis after challenge.

**Figure 7 viruses-15-01873-f007:**
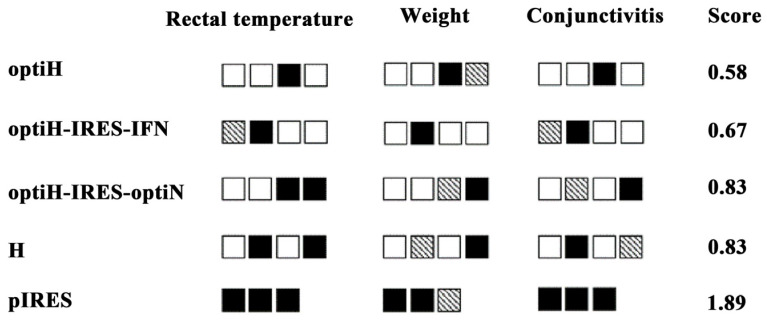
Clinical symptom scores. Rectal temperature (°C), body weight (kg), and conjunctivitis were classified into three levels. Each box represents one animal, and each experimental group contains three or four animals. Black box represents the highest score (2), gray box represents an intermediate score (1), and white box represents the lowest score (0). The scores serve as an indicator of the intensity of clinical symptoms, with higher scores corresponding to more severe symptoms. The specific scoring criteria is detailed in Table 1.

**Figure 8 viruses-15-01873-f008:**
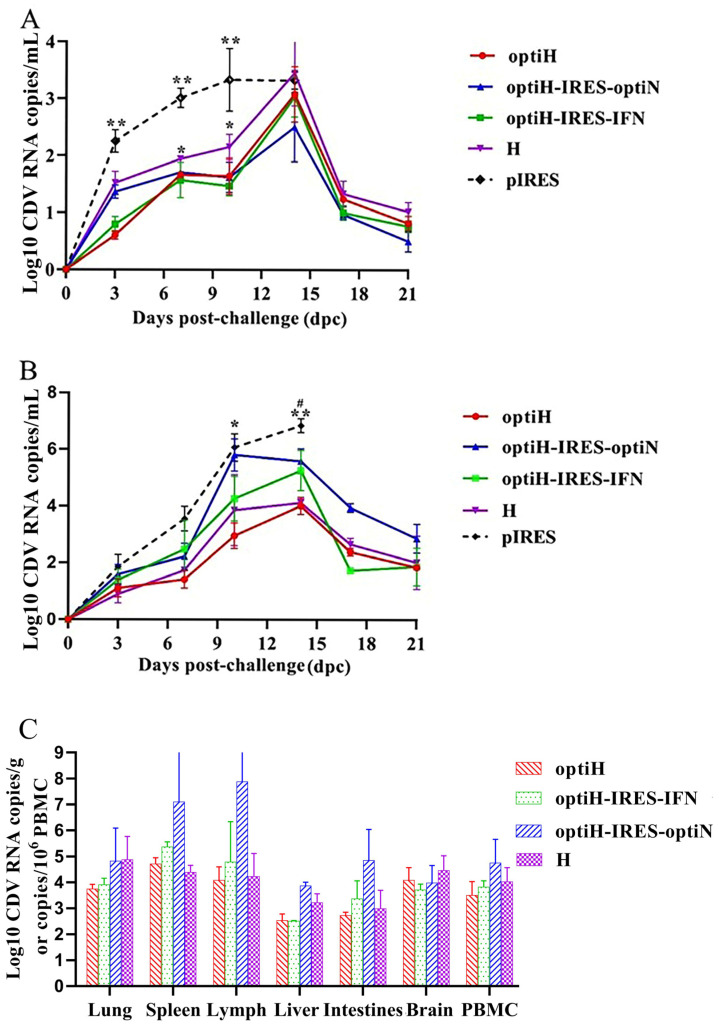
CDV RNA loads in the nasal and rectal swabs and tissues of infected animals using real-time RT-PCR. Both the nasal (**A**) and rectal swabs (**B**) were collected from infected ferrets at various time points. The results show the mean values ± SEM of four ferrets at different time points after the CDV challenge. (**C**) Six tissue types and PBMC were collected and analyzed for CDV RNA loads from infected animals at 21 dpc. The results are presented as mean ± SEM. (* *p* < 0.05, ** *p* < 0.01, compared with the control group; # *p* < 0.05, compared with H).

**Table 1 viruses-15-01873-t001:** Division of ferret immunization group.

Figure Number	DNA Vaccine	Vaccine Dose (µg)	Route
#1, #2, #3, #4	optiH	800/800/500	i.d. + i.m.
#5, #6, #7, #8	optiH-IRES-IFN	800/800/500	i.d. + i.m.
#9, #10, #11, #12	optiH-IRES-optiN	800/800/500	i.d. + i.m.
#13, #14, #15, #16	H	800/800/500	i.d. + i.m.
#17, #18, #19, #20	pIRES	800/800/500	i.d. + i.m.

i.d.: intradermal; i.m.: intramuscular injection.

**Table 2 viruses-15-01873-t002:** Scoring criteria for clinical symptoms in ferret challenge experiments.

Symptom	Clinical Score		
	2	1	0
Conjunctivitis	Severe discharge	Moderate secretion	Mild discharge
Fever	≥39.2 °C and ≥3 days	≥39.2 °C and ≥2 days	37.8–39.2 °C
Weight	Loss > 10%	5% < loss < 10%	Loss ≤ 5%

## Data Availability

The data presented in this study are available in the Appendix A.

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
