# Peer review of "DNA Vaccine Co-Expressing Hemagglutinin and IFN-γ Provides Partial Protection to Ferrets against Lethal Challenge with Canine Distemper Virus"

_viruses, 2023, doi:10.3390/v15091873_

Round 1
Reviewer 1 Report
The authors administered DNA vaccines to ferrets for evaluating the protection against CDV infection, and found that codon optimization of the CDV H gene enhanced immune induction, which was further enhanced by bi-cistronic expression of IFN-g via IRES system. The experiments were properly conducted and directed at their claims. However, the text is not clear and needs to be proofread throughout. In addition, there are several issues that need to be resolved in the notation of the data.
Major points
1. There are many errors in English grammar. Spelling errors are also found as exemplified by Line 41 and 66. Please carefully review the text overall.
2. Figure 3: Nuclear staining is different in each picture, so the pictures should be re-taken. Remake the Figure to directly show the transfected plasmids and the detected antigens on the pictures, which helps the readers to understand what each picture indicates.
3. Figure 3: The numerical count data of the numbers of antigen positive cells per samples are required.
4. Figure A2 and A3: The current pathological data does not show the whole, only a small part of the pathological findings is cut out. Please rework these Figures with a whole view of the tissue and appropriate magnification so that the difference between optiH-IRES-IFN and pIRES can be understood. Please describe the pathological findings more carefully in the text.
5. Figure A3: It is not clear from this Figure whether CDV antigen is stained or not.
6. A plasmid map of the DNA vaccine should be shown to indicate what other elements are in this vaccine besides those listed in Figure 1.
7. The author mentions cell-mediated immunity, but is there any data directly showing that cell-mediated immunity is induced by the vaccines? If not, please refrain from exaggerated descriptions.
8. The sequence alignment of the parental CDV gene and codon optimized gene should be shown.
Minor points
1. Line 175 and 176: Are “cell IL-2” and “cell IL-4” different from normal IL-2 and IL-4?
2. Is there any reference showing that canine ELISA kits can be used for ferret?
3. Figure 4: Please indicate on the Figure or legend what the asterisks mean.
4. Line 245, Results 3.3: Please note in the Results section that the cytokines "in sera" were measured.
5. Line 302, Results 3.5: the "viral shedding" is not an appropriate term and should be replaced with another appropriate term.
6. Fig 8C: Please match the order of bars and bar descriptions.
7. Describe in detail what gene or region of the CDV genome is being amplified for viral RNA detection. Does that amplified region not amplify the DNA vaccine sequence itself?
There are many errors in English grammar. Spelling errors are also found as exemplified by Line 41 and 66. Please carefully review the text overall.
Author Response
Thank you for your comments concerning our manuscript entitled “DNA vaccine co-expressing hemagglutinin and IFN-γ provides partial protection of ferrets against lethal challenge with canine distemper virus”. Those comments are all valuable and very helpful for revising and improving our paper. We have studied comments carefully and have made correction which we hope meet with approval. Revised portion are marked in red in the paper. In addition, we have consulted native English speakers for paper revision before the submission this time. Point-by-point response is included in the response letter.

Reviewer 2 Report
The authors have developed candidate CDV DNA vaccines, and evaluated these in a vaccination and challenge study in ferrets. Three vaccinations resulted in low levels of CDV neutralizing antibodies, and reduced viral loads and clinical signs after lethal CDV challenge infection but did not prevent CDV infection of the brain. The manuscript needs major revision before it can be considered for publication, see the list of major and minor comments below.
Major comments
1. The presentation and interpretation of VN antibodies (Fig 4, results lines 229-240, discussion lines 361-363) contains major errors. The authors claim that VN titers induced by vaccination were > 1:100 (lines 235, 362) but these titers were only measured in week 12, i.e. after CDV challenge. The third vaccination was given in week 6, so the VN antibody levels induced by the full schedule of 3 DNA vaccinations is detected in the samples collected at study week 9. At this time point mean titers ranged between 5 and 15 (Fig. 4).
2. There is another issue with Fig. 4: according to M&M (line 170) the authors used a starting dilution of 1:8. This means that this is also the lower limit of detection of the assay, and it is impossible to detect VN titers below 1:8. The authors should explain how they determined the low mean titers at weeks 3, 6 and 9, I assume these are based on multiple undetectables and a single low positive sample. The figure should be modified to use a logarithmic scale for the y-axis, and should include symbols for the individual measurements per group. The authors should preferably use unique symbols for each of the individual animals in the different treatment groups, and include these in all figures. This way the readers can see which biological responses are associated with protection levels.
3. The discussion is merely a repetition of the results, with the exception of lines 406-416. The authors should discuss their results in the light of the existing literature. A major conclusion should be that the original study goals as formulated in lines 22-23 were not achieved. They cite three papers on DNA vaccination against CDV in their introduction (refs 8-10). How do the data shown here relate to those studies? There are several papers on DNA vaccination against other morbilliviruses, including measles. What can we learn from this? Where do the authors think that DNA vaccines could have a benefit over live-attenuated CDV vaccines? These need a single shot and induce much stronger immune responses. The authors mention maternal antibody interference, but did not test this in the current study. Perhaps DNA vaccines could have a benefit in zoo animals or in protected susceptible wildlife species?
Minor comments
4. Line 23: space between we and constructed
5. Line 41: typo “es”
6. Line 142: superscript 5.0
7. Line 184: the authors report they isolated RNA from blood samples, but do not show data on viremia in Fig. 8. Viremia is a crucial aspect of CDV pathogenesis, and these results should be included in the figure.
Author Response
Thank you for your comments on our manuscript titled "DNA vaccine co-expressing hemagglutinin and IFN-γ provides partial protection of ferrets against lethal challenge with canine distemper virus". These comments helped us improve our manuscript, and provided important guidance for future research.Based on these comments and suggestions, we have made careful modifications on the original manuscript. All changes made to the text are in red color.

Round 2
Reviewer 1 Report
The manuscript has been adequately revised.
Author Response
Thank you for your review.
Reviewer 2 Report
The authors have addressed most of the comments of the reviewer. However, they should still modify the abstract to reflect that the vaccine was poorly immunogenic. The current statement in lines 31/32 is not compatible with that in lines 396/397. Moreover, the statement in lines 399/400 is incorrect as in each group of vaccinated animals at least one animal died as a result of challenge infection.
NA
Author Response
Thank you for reviewing my manuscript again. We have carefully revised your question, and we have added a specific time description to the description of vaccine immune effect in the abstract and results, which is helpful to judge the vaccine effect more intuitively. In addition, we have made language-based edits to the manuscript in order to improve its readability while preserving its original meaning. Change the part to be displayed in red in the manuscript.
